

# A comparison of antibiotic resistance genes and mobile genetic elements in wild and captive Himalayan vultures

Jundie Zhai[1,2], You Wang[1,2], Boyu Tang[1,2], Sisi Zheng[3], Shunfu He[4], Wenxin Zhao[4], Jun Lin[2], Feng Li[2], Yuzi Bao[2], Zhuoma Lancuo[5], Chuanfa Liu[6] and Wen Wang[1]

[1] State Key Laboratory of Plateau Ecology and Agriculture, Qinghai University, Xining, China
[2] College of Eco-Environmental Engineering, Qinghai University, Xining, China
[3] Animal Disease Prevention and Control Center of Qinghai Province, Xining, China
[4] Xining Wildlife Park of Qinghai Province, Xining, China
[5] College of Finance and Economics, Qinghai University, Xining, China
[6] College of Life Sciences, University of Chinese Academy of Sciences, Beijing, China

Corresponding authors
Chuanfa Liu,
liuchuanfa15@mails.ucas.ac.cn
Wen Wang, 007cell@163.com

## ABSTRACT

As the most widely distributed scavenger birds on the Qinghai-Tibetan Plateau, Himalayan vultures (*Gyps himalayensis*) feed on the carcasses of various wild and domestic animals, facing the dual selection pressure of pathogens and antibiotics and are suitable biological sentinel species for monitoring antibiotic resistance genes (ARGs). This study used metagenomic sequencing to comparatively investigate the ARGs and mobile genetic elements (MGEs) of wild and captive Himalayan vultures. Overall, the resistome of Himalayan vultures contained 414 ARG subtypes resistant to 20 ARG types, with abundances ranging from 0.01 to 1,493.60 ppm. The most abundant resistance type was beta-lactam (175 subtypes), followed by multidrug resistance genes with 68 subtypes. Decreases in the abundance of macrolide-lincosamide-streptogramin (MLS) resistance genes were observed in the wild group compared with the zoo group. A total of 75 genera (five phyla) of bacteria were predicted to be the hosts of ARGs in Himalayan vultures, and the clinical (102 ARGs) and high-risk ARGs (35 Rank I and 56 Rank II ARGs) were also analyzed. Among these ARGs, twenty-two clinical ARGs, nine Rank I ARG subtypes, sixteen Rank II ARG subtypes were found to differ significantly between the two groups. Five types of MGEs (128 subtypes) were found in Himalayan vultures. Plasmids (62 subtypes) and transposases (44 subtypes) were found to be the main MGE types. Efflux pump and antibiotic deactivation were the main resistance mechanisms of ARGs in Himalayan vultures. Decreases in the abundance of cellular protection were identified in wild Himalayan vultures compared with the captive Himalayan vultures. Procrustes analysis and the co-occurrence networks analysis revealed different patterns of correlations among gut microbes, ARGs, and MGEs in wild and captive Himalayan vultures. This study is the first step in describing the characterization of the ARGs in the gut of Himalayan vultures and highlights the need to pay more attention to scavenging birds.

## INTRODUCTION

Antibiotic resistance (AR) has become a major global issue, posing a threat to human and animal health, food safety, and ecological security (*Hernando-Amado et al., 2019*). Increases in antibiotic production, overuse, and abuse in various settings are related to the spread of antibiotic-resistance bacteria (ARB) and their antibiotic-resistance genes (ARGs) in different environments (*Allen et al., 2010*). ARGs and their corresponding carrier ARB are global emerging environmental contaminants (*Pruden et al., 2006*). Moreover, under antibiotic pressure, ARGs can be transferred to bacteria of the same or other species and genera through horizontal gene transfer (HGT) mediated by mobile genetic elements (MGEs), thereby accelerating the development and persistence of antibiotic resistance (*Martínez, Coque & Baquero, 2015*). Increasing evidence shows that environmental and commensal bacteria comprise a vast and largely unexplored reservoir of ARGs (*Caniça et al., 2015*; *Li et al., 2020*). The origins, detection, fate, and health risks of ARGs have been the focus of research, while most studies tend to focus on diverse environmental compartments (*e.g.*, the soil system *Chen et al., 2021*), aquatic systems (*Tang et al., 2023*), air system (*Gwenzi et al., 2022*), and livestock system (*Ma, McAllister & Guan, 2021*). ARGs related to wild animals are much less well-recognized and still in infancy. Increasing urbanization and fragmentation of natural habitats may occasionally expand the role of wild animals in the spread of ARGs because wildlife increasingly feed on food and water contaminated by residual ARGs and ARB in their living environments (*Kumari et al., 2022*). The activities of some migratory wild animals can mediate the transfer of ARGs to remote areas and further aggravate environmental dissemination.

Compared with other non-flying wild vertebrates, birds can fly long distances and have been gradually recognized as important sentinels, reservoirs, and potential spreaders for disseminating environmental ARGs across various ecosystems (*Wang et al., 2017*). Billions of birds migrate between their wintering and breeding grounds each year, and their migration flyways cover all continents, including Antarctica, which would accelerate the globalization of ARGs and ARB (*Hernando-Amado et al., 2019*). Several studies have confirmed ARGs and ARB from wild birds, including waterbirds (*e.g.*, *Anser erythropus Liu, Xu & Feng, 2023*, and *Larus argentatus Ruzauskas & Vaskeviciute, 2016*), *Passer domesticus* (*Dolejská et al., 2008*), and *Corvidae* birds (*Oravcová et al., 2018*). Trillions of microbes live in the guts of birds, which are essential for maintaining host health, and can also function as reservoirs of ARGs and ARB because ARGs may quickly colonize in their gut because of contaminated food or water (*Cao et al., 2020*). The gut microbiota of birds fluctuates and is influenced by host phylogeny, diet, and other environmental factors, resulting in diverse and dynamic ARG profiles (*Luo et al., 2022*). A study has shown that wild birds living in environments affected by human activities usually carry more ARGs than wild birds in remote areas (*Miller et al., 2020*). Therefore, given that zoos are places where tourists congregate, it becomes particularly important to conduct an in-depth investigation into the ARB and ARGs carried by birds within the zoo. This aspect is also a key area of focus within One Health.

Several studies have investigated the distribution of ARGs across various environments (soils, surface waters, and the atmosphere) of the Qinghai-Tibet Plateau (*Wang et al., 2022*). Wild and domestic animals in the Qinghai-Tibetan Plateau area were also widely studied to understand the patterns of ARGs dissemination (*Cao et al., 2020*; *Lin et al., 2020*; *Zhao et al., 2023*). Himalayan vultures (*Gyps himalayensis*) are distributed mainly in the Qinghai-Tibetan Plateau, where they feed on the rotting plateau domestic animals (*e.g.*, yaks (*Bos mutus*), Tibetan sheep (*Ovis aries*), domestic dogs, and domestic horses) and other plateau wildlife carcasses. We hypothesize that ARB and ARGs will also gather in the guts of Himalayan vultures through the saprophagous food chain, especially by eating many livestock carcasses. However, the gut resistome of this special bird species is still unknown. Furthermore, the published gut microbiome research has shown that most pathogenic bacteria exist in the guts of Himalayan vultures (*Meng et al., 2017*). Thus, the spread of ARGs among these gut-pathogenic bacteria may constitute a serious public health problem. As a national second-class protected bird species in China, captive breeding has developed into one of the effective measures to protect Himalayan vultures. Xining Wildlife Park is the only zoo in China that can artificially raise and breed Himalayan vultures. However, it is still unclear whether distinct ARGs exist in Himalayan vultures living in wild and human-created environments and how environmental changes shape the gut resistomes of Himalayan vultures. In the present study, 19 fecal samples, including eight fecal samples from wild Himalayan vultures and 11 fecal samples from captive Himalayan vultures, were collected and sequenced using metagenomic methods, and their ARGs, MGEs, and gut microbes were detected and compared to determine the ARGs profiles in Himalayan vultures.

## MATERIALS AND METHODS

### Ethics statement

This study conformed to the guidelines for the care and use of experimental animals established by the Ministry of Science and Technology of the People's Republic of China (Approval number: 2006-398). The research protocol was reviewed and approved by the Ethical Committee of Qinghai University. This study did not involve the capture, direct manipulation, or disturbance of Himalayan vultures.

### Samples collection

Nineteen fresh fecal samples of Himalayan vultures were collected from wild and captive individuals. Among them, eight wild fecal samples (Wild group) were randomly selected during the field survey of Himalayan vultures in Yushu City, Qinghai Province, China. All wild Himalayan vultures were considered healthy based on physical appearance and behavior through field observation. Eleven fecal samples (Zoo group) were opportunistically collected from captive Himalayan vultures reared in Xining Wildlife Park, China. All captive individuals were not given antibiotics or other medicines for half a year before this study. All feces were sampled immediately after defecation and transported to the laboratory using liquid nitrogen.

## DNA extraction and metagenomic sequencing

Genomic DNA was extracted from all fecal samples using the Qiagen QIAamp DNA Stool Mini Kit (Qiagen, Hilden, Germany) according to the manufacturer's instructions. Extracts were treated with DNase-free RNase to eliminate RNA contamination. DNA concentrations were measured on the Qubit 2.0 fluorimeter (Invitrogen, USA). DNA purity was determined with Nanodrop (Thermo Scientific, Waltham, MA, USA) by measuring the 260/280 and 260/230 absorbance ratios. A paired-end (PE) library with an insert size of 350 bp for each sample was constructed, followed through high-throughput sequencing using the BGISEQ-500 sequencer with PE reads of length $2 \times 150$ bp.

## Raw data processing and taxonomy profiling

The raw sequencing reads from each sample were independently processed for quality inspection. Initial data quality control was performed using the FastQC (https://www.bioinformatics.babraham.ac.uk/projects/fastqc), and then reads were filtered and trimmed using Trimmomatic (*Bolger, Lohse & Usadel, 2014*) with the following parameters: ILLUMINACLIP:adapters.fa:2:30:10, SLIDINGWINDOW:4:15, MINLEN:75. Host DNA was removed using paired-end mapping with Bowtie2 (version 2.4.5) (*Langmead & Salzberg, 2012*) against a Himalayan vulture reference genome GWHBAOP00000000 (http://bigd.big.ac.cn/gwh). Taxonomic profiles were generated from high-quality reads using kraken2 (*Lu & Salzberg, 2020*), and bracken was used to estimate the abundance of microbial taxonomic profiles.

## ARGs and MGEs annotation and quantification

Based on the high-quality metagenomic reads, the ARGs were identified and classified into different types and subtypes using the ARGs-OAP pipeline 2.0 against the SARG database (the structured ARG reference database) (*Yin et al., 2018*). ARGs were annotated according to the following criteria: an e-value cutoff of 1e−7, amino acid similarity of 80%, and 75% hit length. The ARGs between the two groups were compared at the type and subtype level with the "ppm" unit (one read per million reads).

For MGE annotation, the high-quality reads obtained were compared to the MGE database (*Ellabaan et al., 2021*) using the ARGs-OAP pipeline 2.0. A read was identified as a MGE, including various types such as plasmids, integrons, transposases, insertion sequence transposases (*ist*), and insertion sequence (*IS*) genes, by using BLASTX with an e-value threshold of 1e−7 and a minimum amino acid similarity of 80%. Finally, the MGE abundances were also normalized with the "ppm" unit.

## Host identification analysis

These high-quality metagenomic reads were assembled into contigs using MEGAHIT (*Li et al., 2015*). Contigs with a length ≥ 500 bp were selected as the final assembling result. The open reading frame (ORF) was identified using METAProdigal (*Hyatt et al., 2012*), and the non-redundant gene catalog was then constructed using CD-HIT (v4.6.1) (*Li & Godzik, 2006*) with 90% sequence identity and 90% coverage. Salmon software (*Patro et al., 2017*) was used to calculate the abundance of each non-redundant gene. BLASTX was used for

ARG annotation by matching the non-redundant gene sets with the SARG database, and then the NCBI-NR database was used to classify the annotated ARG contigs.

## Identification of clinical and high-risk ARGs

Clinical ARGs were considered to be ARGs that occur in human pathogens. Bacterial genomes associated with human disease were retrieved from the PATRIC (Pathosystems Resource Integration Center) database (https://www.bv-brc.org/) and combined with the SARG database to annotate clinical ARGs. We considered hits with identity $\geq$ 90%, an alignment length of $\geq$ 75 bp, and an e-value cutoff of 1e−7 as the clinical ARGs from our high-quality metagenomic reads. The risk of ARGs was analyzed according to the omics-based framework with three criteria: (i) anthropogenic enrichment, (ii) mobility, and (iii) host pathogenicity (*Zhang et al., 2021*). These human-associated, mobile ARGs were considered high-risk ARGs, further divided into Rank I ARGs (meeting all three criteria, defined as current threats) and Rank II ARGs (meeting criteria (i) and (ii), defined as future threats). Therefore, the Rank I and Rank II ARGs were screened from our data to assess health risks.

## Statistical analysis

All statistics were performed using R software version 4.3.1 (*R Core Team, 2020*). Statistical comparisons were performed using nonparametric Wilcoxon tests between the wild and zoo groups. The multiple test correction was conducted using Bonferroni correction. For all statistical tests, a $P$ value of less than 0.05 was considered statistically significant. Principal Coordinate Analysis (PCoA) based on the Bray-Curtis distance with permutational multivariate analysis (Adonis) was performed to show the distinction of ARGs, MGEs, and gut microbes between two groups using the "vegan" package. Procrustes analyses were also performed with the "vegan" package. Venn diagrams, Bar charts, and scatter diagrams were produced using the "ggplot2" package. The network analysis was conducted based on Spearman correlation with a correlation coefficient >0.9, and $P$ value <0.05, and network diagrams were produced using the "ggraph" package.

## RESULTS

### The profiles of ARGs in Himalayan vultures

A total of 20 ARG types, which consisted of 414 ARG subtypes, were identified in all samples (Fig. 1A). In detail, the study detected resistance genes for beta-lactam, multidrug, tetracycline, aminoglycoside, macrolide-lincosamide-streptogramin (MLS), vancomycin, polymycin, trimethoprim, chloramphenicol, and quinolone. The number of ARG subtypes varied across samples, ranging from 93 to 237. The most abundant resistance type was beta-lactam (175 subtypes), followed by multidrug resistance genes with 68 subtypes. Tetracycline, aminoglycoside, and MLS ranked third to fifth, with 36, 30, and 26 subtypes, respectively. The average abundances of the ARG types ranged from 0.01 to 1,493.60 ppm between the wild and zoo groups (Fig. 1B). Tetracycline was the most abundant ARG type (1,394.91 ppm), followed by beta-lactam (524.02 ppm), multidrug (483.00 ppm), MLS (84.91 ppm), bacitracin (45.34 ppm), and aminoglycoside (32.08 ppm). Tetracenomycin

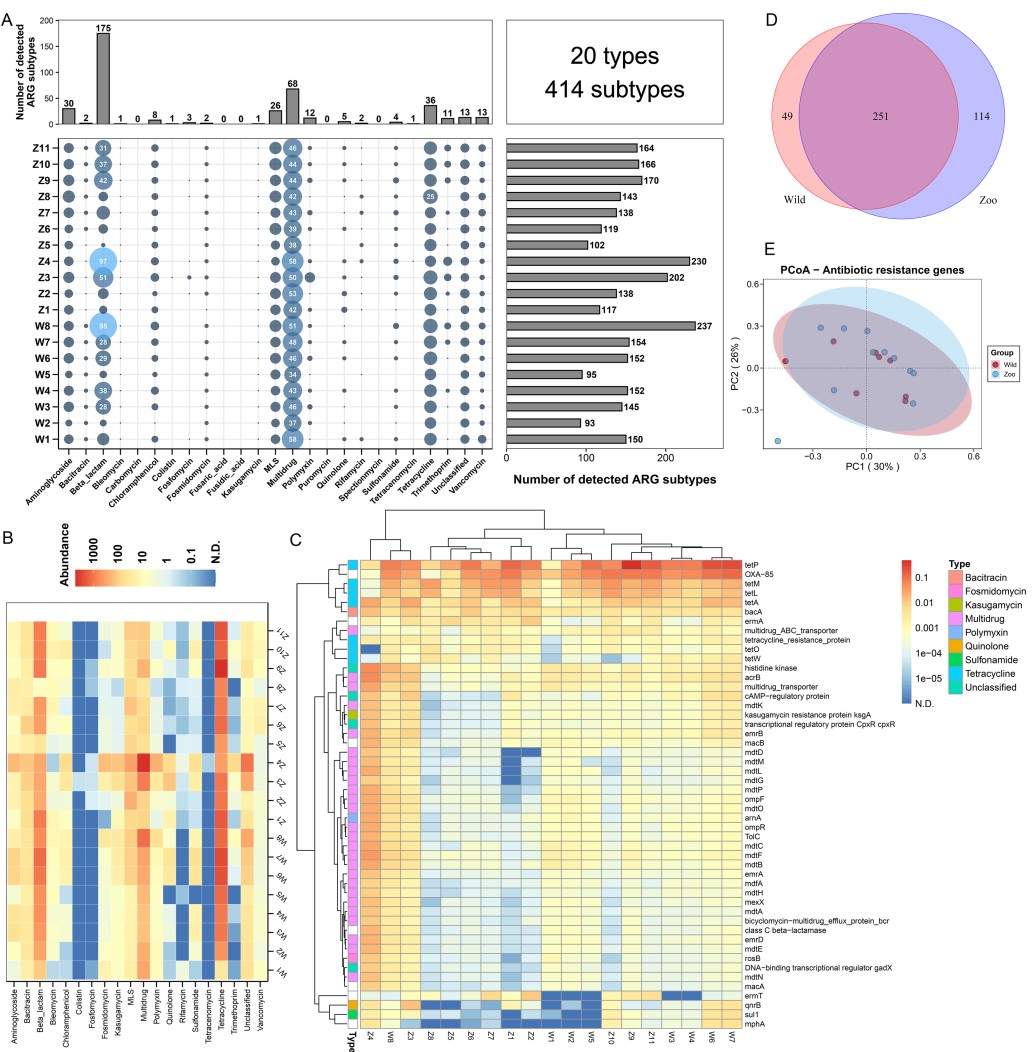

**Figure 1  Antibiotic resistance genes (ARGs) in Himalayan vultures.** (A) Statistical combination map of ARGs types and subtypes in each sample. (B) Heatmap of the top 50 ARGs type abundances in each sample. (C) Heatmap of the ARGs subtype abundances in each sample. (D) Venn diagrams of the numbers of core ARGs in both wild and zoo groups. (E) Principal coordinate analysis (PCoA) plots based on Bray–Curtis distances at the level of ARGs subtypes.

was the least abundant type. Decreases in the abundance of MLS resistance genes ($P < 0.05$) were observed in the wild group compared with the zoo group. However, the abundances of the other ARGs were not different between the wild and zoo groups. Among ARG subtypes, the tetracycline resistance gene *tetP* was the dominant ARG subtype (950.22 ppm), followed by the beta-lactam, multidrug, and bacitracin resistance genes *OXA-85* (488.68 ppm), *acrB* (53.93 ppm), and *bacA* (45.21 ppm) (Fig. 1C). Furthermore, 251 core ARG subtypes were shared by both groups, with 49 unique subtypes in the wild group and 114 unique subtypes in the zoo group (Fig. 1D). Principal coordinates analysis was

performed to cluster samples using the relative abundance of the ARGs. We observed no significant difference between the groups (Adonis, $R^2 = 0.078$, $P = 0.187$; Fig. 1E).

## The profiles of MGEs in Himalayan vultures

To explore the potential mechanism of ARG dissemination in Himalayan vultures, we characterized the number and abundance of MGEs in both wild and zoo groups. Five MGEs types (128 subtypes) were identified in both groups (Fig. 2A). Plasmids (62 subtypes) and transposases (44 subtypes) were found to be the main MGE types. The number of MGE subtypes varied across samples, ranging from 29 to 81. The total number of MGE subtypes detected in the zoo group was significantly higher than that in the wild group (Fig. 2B). For the two groups, *rep22* (46.64 ppm), *tnpA* (1,018.75 ppm), *IS91* (32.86 ppm), *Int-Tn916* (109.50 ppm), and *istA2* (1.81 ppm) were found to be the dominant subtypes belonging to plasmid, transposase, IS, integrase, and *ist*, respectively (Fig. 2C). The shared and unique MGEs were identified in the wild and zoo groups. The study detected 73 core MGEs, 10 unique MGE subtypes in the wild group, and 45 unique subtypes in the zoo group (Fig. 2D). Furthermore, PCoA demonstrated that the MGEs were not significantly (Adonis, $R^2 = 0.049$, $P = 0.464$; Fig. 2E) separated between the two groups.

## Microbial community compositions of Himalayan vultures

Microbial community analysis was performed to identify potential ARG and MGE carriers using metagenomic sequencing. Shotgun sequencing resulted in 5.28–13.15 Gb of raw reads per sample. After quality control, each sample contained about 0.85−5.82 Gb high-quality reads (Table S1). The taxonomic compositions were detected at the phylum and genus levels. A total of 41 phyla were identified in both groups. Fusobacteria (50.58%) was the dominant phylum, followed by Proteobacteria (24.84%), Firmicutes (18.04%), Bacteroidetes (3.03%), and Actinobacteria (2.03%) (Fig. 3A). The relative abundance of these top five phyla occupied 98.53% of the total sequences. A total of 1,451 genera were identified in both groups. *Fusobacterium* (52.33%) was dominant, followed by *Plesiomonas* (12.73%), *Clostridium* (10.00%), *Escherichia* (6.36%), and *Streptococcus* (0.92%) (Fig. 3B). The relative abundance of the top five genera occupied 82.02% of total sequences. PCoA analysis was used to estimate the microbial structure and diversity differences among samples. The microbial community structures were not significantly different between the two groups (Adonis, $R^2 = 0.077$, $P = 0.134$; Fig. 3C).

## Correlations and co-occurrence patterns among gut microbes, ARGs, and MGEs

In the wild group, Procrustes analysis (Figs. 4A–4C) revealed significant correlations between the ARGs and MGEs ($r = 0.33597$, $P = 0.004$), the ARGs and microbes ($r = 0.39424$, $P = 0.006$), and the microbes and MGEs ($r = 0.29398$, $P = 0.002$). The ARGs, MGEs, and gut microbes correlated significantly with each other, suggesting that the microbial community composition structured the gut ARGs and MGEs composition in the fecal microbiome of wild Himalayan vultures. In the zoo group, Procrustes analysis (Figs. 4D–4F) revealed a significant correlation between the ARGs and MGEs ($r = 0.33908$, $P = 0.001$), while the correlations between the ARGs and microbes ($r = 0.77790$,

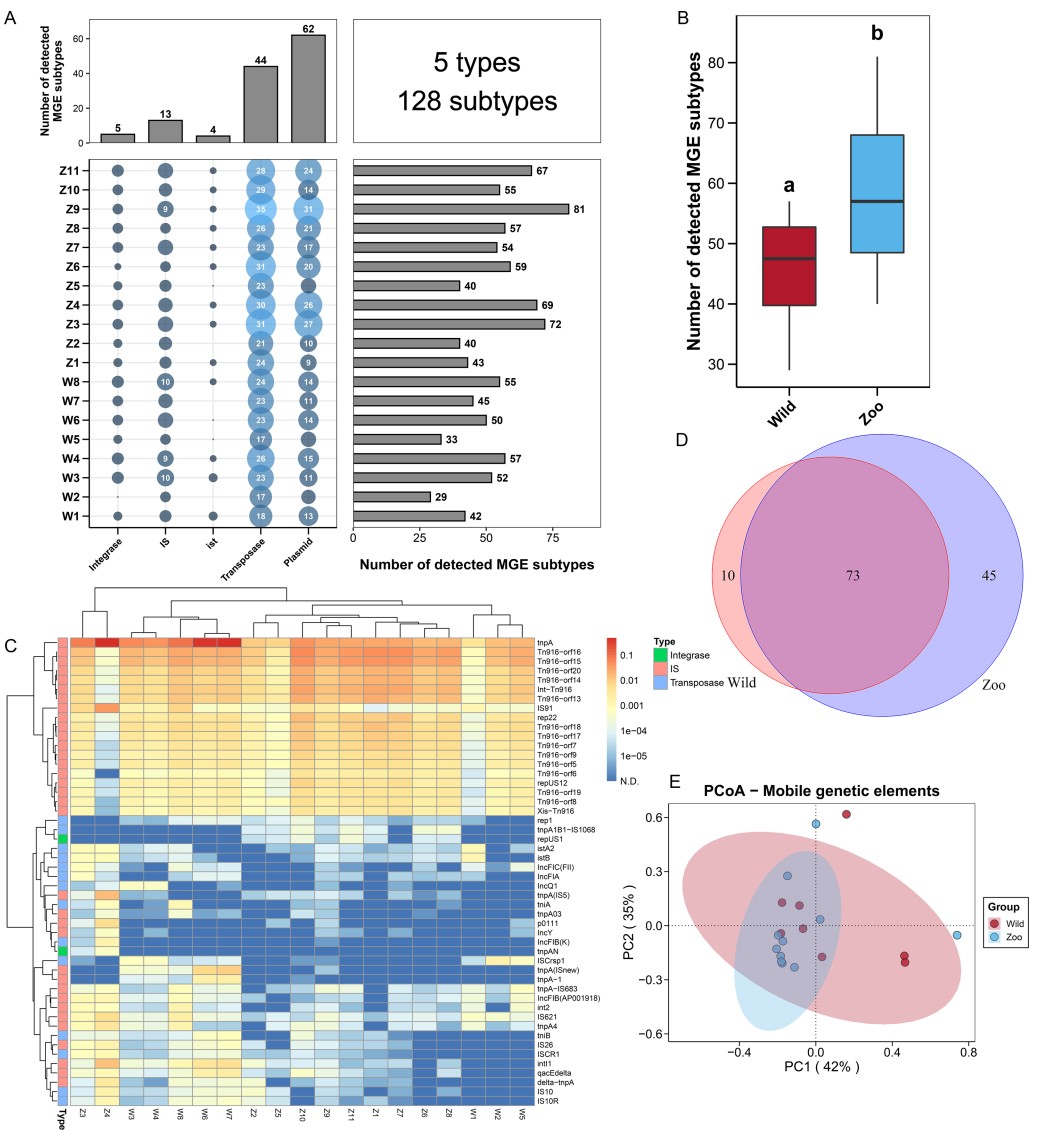

**Figure 2  Mobile genetic elements (MGEs) in Himalayan vultures.** (A) Statistical combination map of MGEs types and subtypes in each sample. (B) Comparison of the total MGEs numbers in each group. (C) Heatmap of the top 50 MGEs type abundances in each sample. (D) Venn diagrams of the numbers of core MGEs in both wild and zoo groups. (E) Principal coordinate analysis (PCoA) plots based on Bray–Curtis distances at the level of MGE subtypes.

$P = 0.206$), and the microbes and MGEs ($r = 0.71092$, $P = 0.079$) were not significant. It was inferred that the antibiotic resistome became decoupled from the microbiome of the captive Himalayan vultures.

The co-occurrence patterns were used to detect the transmission route of ARGs in Himalayan vultures (Fig. 5). In wild Himalayan vultures, 108 ARGs and 52 MGEs showed high co-occurrence with 54 gut microbial genera (Fig. 5A). Among these co-occurrence linkages, *tnpA*, *IS10*, *IS10R*, and *IS26* were the top four highly correlated degree MGEs,

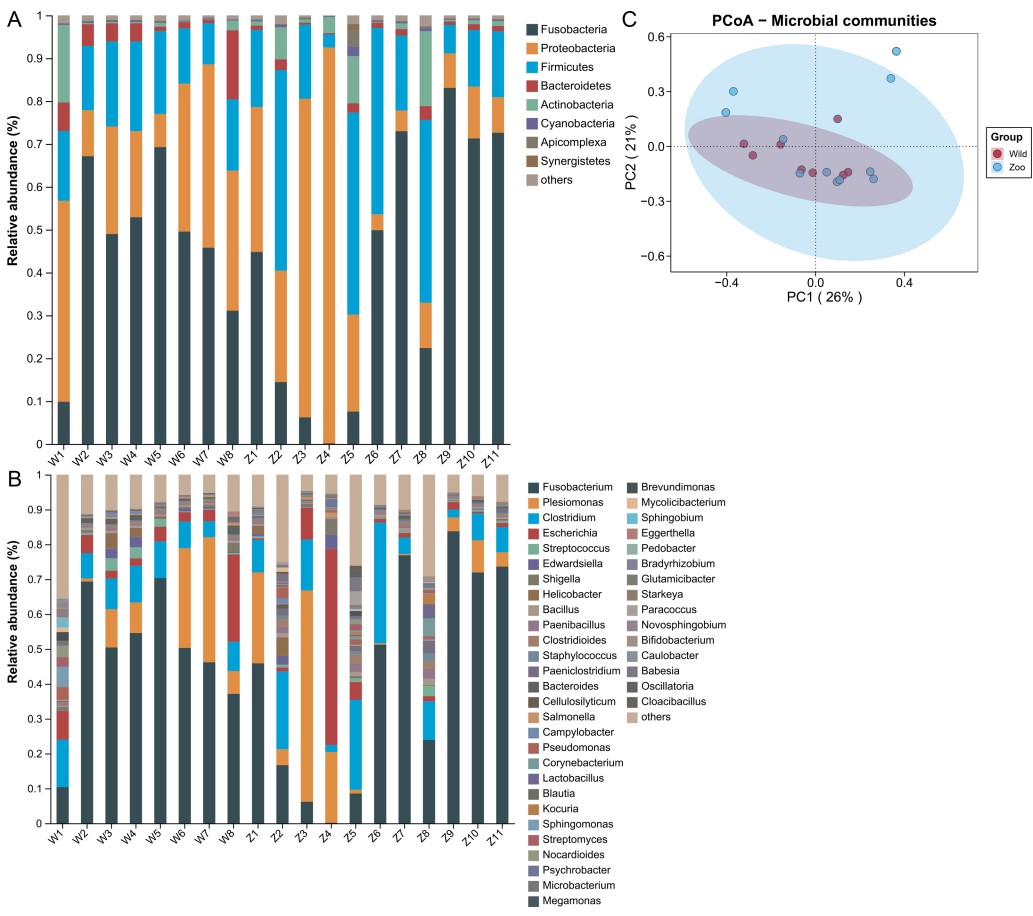

**Figure 3 Microbial community composition of Himalayan vultures.** Taxonomic analyses at the levels of (A) phylum and (B) genus. (C) Principal coordinate analysis (PCoA) plots based on Bray–Curtis distances at the level of genera.

which linked most ARGs to target bacterial genera. In captive Himalayan vultures, 114 ARGs and 67 MGEs exhibited high co-occurrence with 47 gut microbial genera (Fig. 5B). The top five MGEs linked most ARGs to target bacterial genera were *tnpAN*, *IncFIB(K)*, *p0111*, *ColpVC*, and *tnpA10*.

## Identification of bacterial hosts of ARGs in Himalayan vultures

Gut microbes of wildlife were considered an important reservoir of ARGs. A total of 712,303 contigs were assembled, with filtering the length of the contigs <500 bp (Table S2). In total, 75 genera (five phyla) were explored to be the hosts of ARGs in Himalayan vultures. As illustrated in Fig. 6A, *Fusobacterium* (35.41%), *Cetobacterium* (17.41%), *Clostridium* (14.92%), and *Escherichia* (13.51%) were the dominant ARG carriers in the wild group, with *Fusobacterium* carrying beta-lactam, *Cetobacterium* and *Clostridium* carrying tetracycline, and *Escherichia* carrying multidrug. In the zoo group (Fig. 6B), *Fusobacterium* (24.84%), *Escherichia* (22.44%), *Clostridium* (14.82%), and *Cetobacterium* (14.46%) were also the dominant ARG carriers. These results indicated that an ARG could come from more than

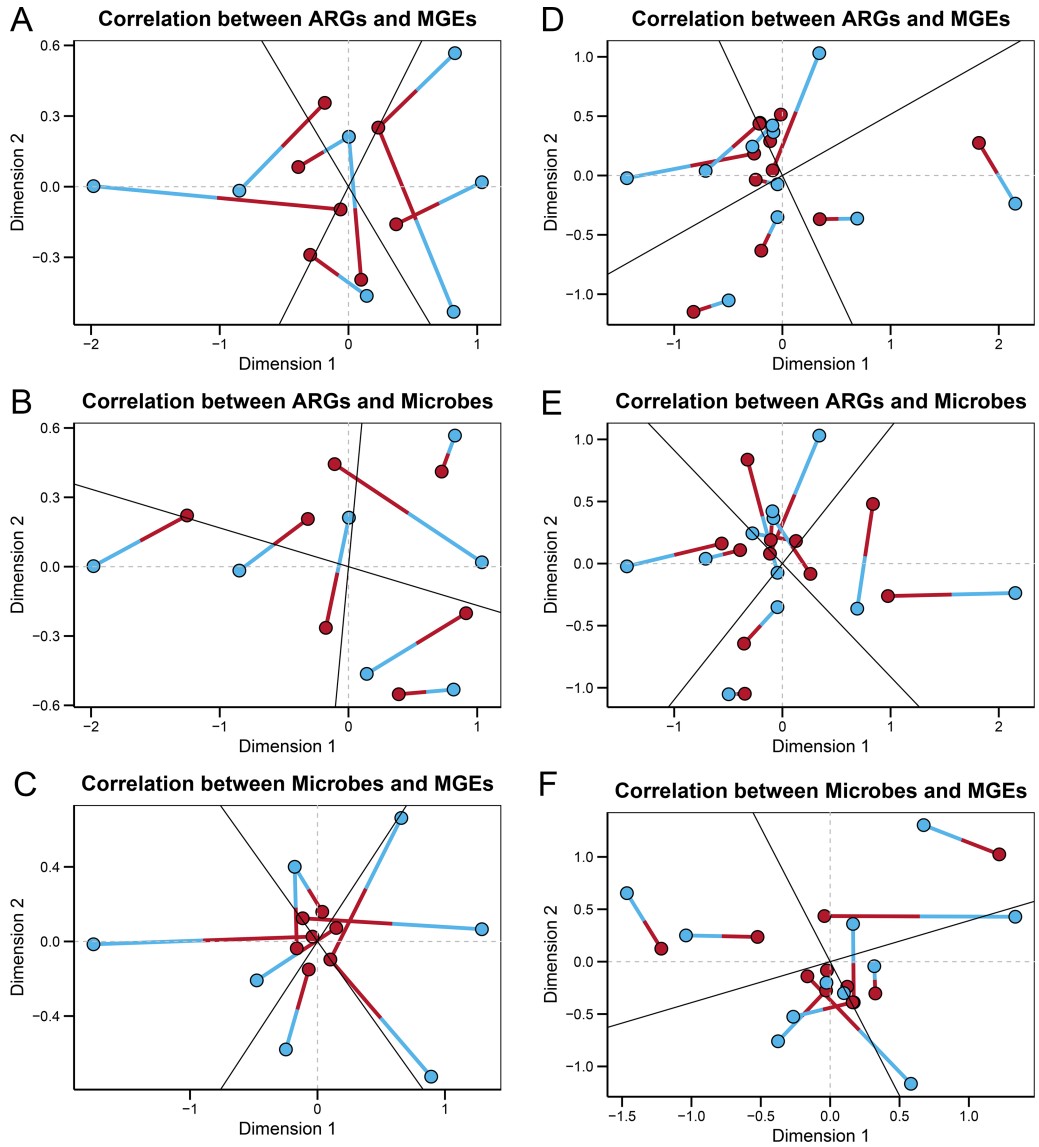

**Figure 4 Relationships among gut microbes, antibiotic resistance genes (ARGs) and mobile genetical elements (MGEs).** Procrustes analyses reveals the correlations between the (A) ARGs and MGEs, (B) ARGs and gut microbes, and (C) gut microbes and MGEs in the wild group. Procrustes analyses reveals the correlations between the (D) ARGs and MGEs, (E) ARGs and gut microbes, and (F) gut microbes and MGEs in the zoo group. In the results, points with different colors represent different analysis data, and the two points connected by a line are from the same sample.

one carrier and that one host could carry more than one ARG. Furthermore, the results further indicated that the main carriers of ARGs in Himalayan vultures were harmful bacteria or pathogens, such as *Fusobacterium*, *Clostridium*, and *Escherichia*.

## Resistance mechanisms of ARGs in Himalayan vultures

Most ARGs in Himalayan vultures were classified as antibiotic deactivation (223 ARGs), efflux pump (80 ARGs), and cellular protection (69 ARGs) mechanisms (Fig. 7A).
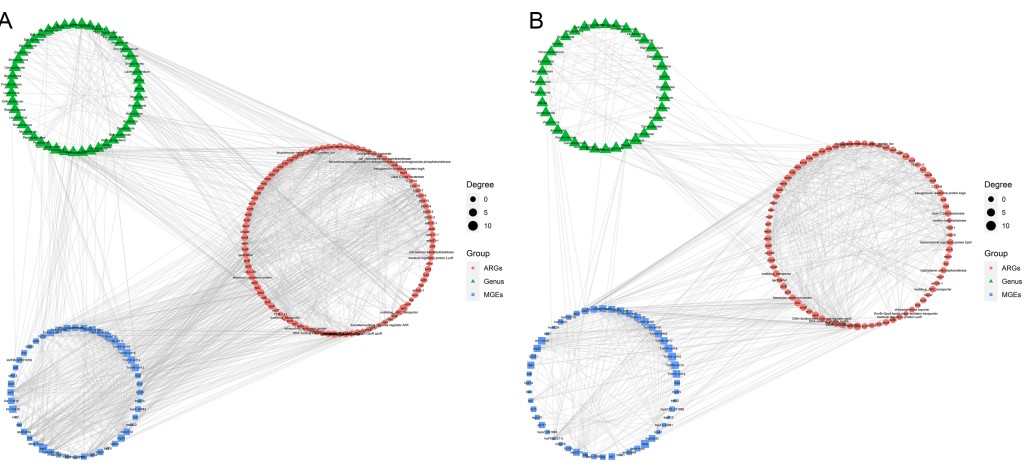

**Figure 5  Network analysis.** Network analysis reveals the co-occurrence relationships among antibiotic resistance genes (ARGs), mobile genetic elements (MGEs) and gut microbes (at the genus level) in (A) wild and (B) zoo groups.

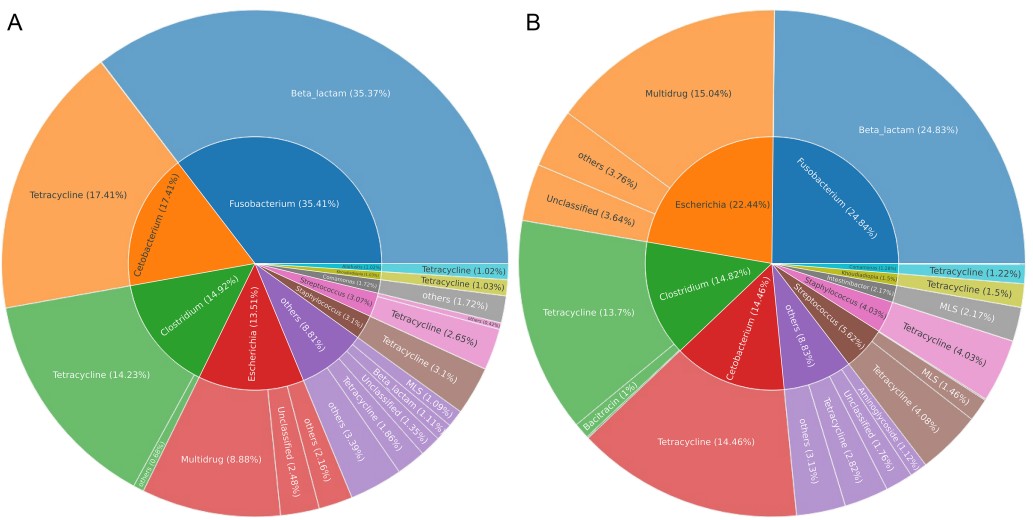

**Figure 6  The antibiotic resistance genes (ARGs) hosts in (A) wild and (B) zoo Himalayan vultures.** The antibiotic resistance genes (ARGs) hosts in (A) wild and (B) zoo Himalayan vultures. The inner circle shows the annotation of the ARGs host at the genus level. The outer circle shows the composition of the ARG types.

Approximately 60% of ARGs were from the efflux pump, while 21% were from the antibiotic deactivation, cellular protection (14%), and regulators (2%) (Fig. 7A). Additionally, decreases in the abundance of cellular protection were identified ($P < 0.05$) in wild Himalayan vultures compared with the captive Himalayan vultures, while other resistance mechanisms did not change significantly between the two groups (Fig. 7B).

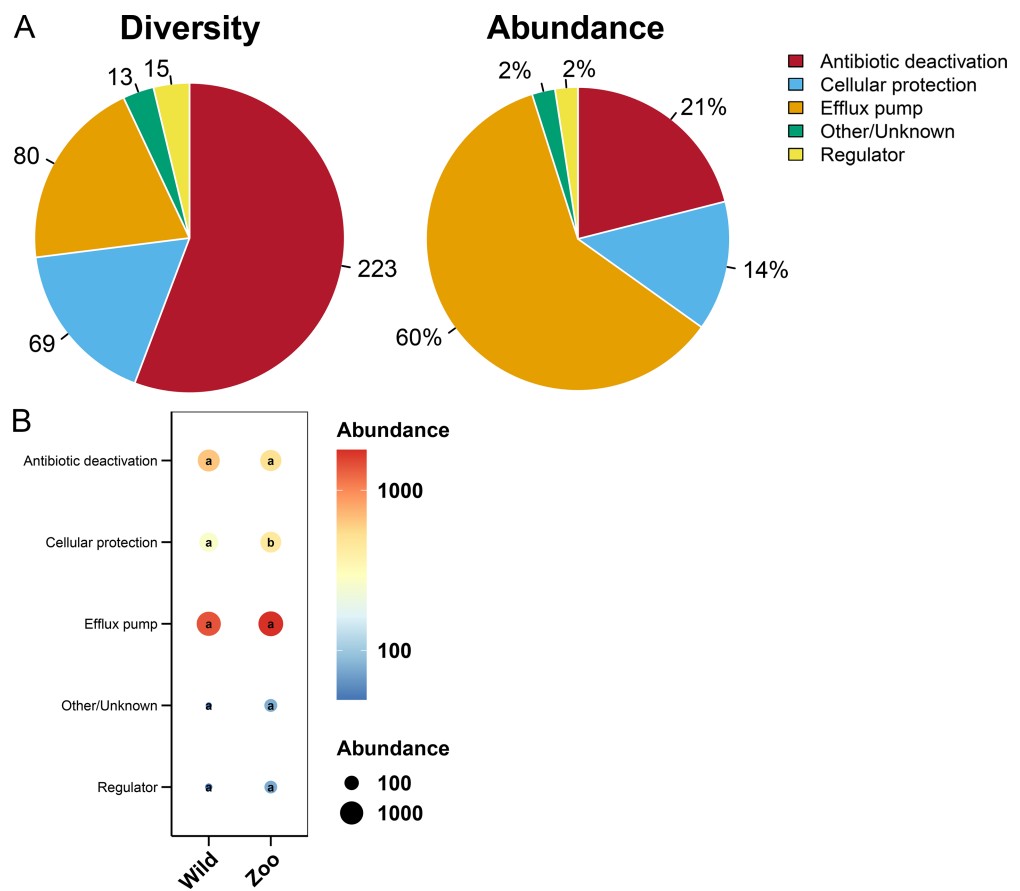

**Figure 7** **Resistance mechanisms in Himalayan vultures.** (A) Diversity and abundance of resistance mechanisms. (B) Comparison of the resistance mechanisms abundances in each group. Values with different lowercase letters between groups were significantly different ($P < 0.05$).

## Detection of clinical and high-risk ARGs in Himalayan vultures

To evaluate the relationships between the ARGs of Himalayan vultures and clinically relevant ARGs, we detected 102 clinical ARGs in Himalayan vultures (Table S3). Among them, abundances of *tetQ*, *tetL*, *erm*, *tetP*, and *tetM_1* were the top five dominant clinical ARGs. Twenty-two clinical ARGs were found to differ significantly between the two groups. We also used an omics-based framework to identify those high-risk ARGs that posed significant potential risks to public health. Overall, 35 and 56 ARG subtypes were identified as current (Rank I) and future (Rank II) threats, respectively. The top five types of 35 Rank I ARGs were aminoglycoside, beta-lactam, MLS, tetracycline, and sulfonamide (Table S4). Nine ARG subtypes were found to differ significantly between the two groups ($P < 0.05$), including the higher abundances of *blaA*, *CfxA2*, and *MOX-7* in the wild group and the higher abundances of *aadA*, *arr*, *erm(TR)*, *ermT*, *lnuB*, and *tetS* in the zoo group. The top five types of 56 Rank II ARGs included multidrug, bacitracin, tetracycline, aminoglycoside, and MLS (Table S5). Sixteen ARG subtypes were found to differ significantly between the two groups. Among them, higher abundances of *aac(6′)-I*, *acrA*, *qnrB*, and *tet40*

were identified in the wild group than in the zoo group, while the zoo group had higher abundances of *acrB*, *aph(3′)-I*, *cep*, *emrB*, *H-NS*, *lnuA*, *mdtC*, *mdtD*, *mdtL*, *mdtO*, *mdtP*, and *tetA* than the wild group.

## DISCUSSION

Because of their unique scavenging behavior, Himalayan vultures are exposed to many pathogens from rotting foods (*Meng et al., 2017*) and face antibiotic selection pressure from dead domestic animals. Based on this, we hypothesize that the ecological niche and feeding behavior of Himalayan vultures may make them highly susceptible to the spread of antibiotic-resistant pathogens. However, data on ARGs related to Himalayan vultures remain scarce. For the first time, we performed metagenomics analysis to describe and compare the characterization of the ARGs and MGEs in the gut microbiomes of both wild and captive-reared Himalayan vultures.

Overall, the resistome of Himalayan vultures contained 414 ARG subtypes resistant to 20 ARG types, with abundances ranging from 0.01 to 1,493.60 ppm, and showed some differences between the two groups. The ARG types detected in Himalayan vultures were also commonly identified in other bird species, such as waterfowl and gulls (*Cao et al., 2020*; *Miller et al., 2020*; *Lin et al., 2020*). The results demonstrated that the ARG subtypes in the Himalayan vulture gut microbiomes conferred resistance to almost all major antibiotic types commonly used in clinical and agricultural practice. Among the most common were genes conferring resistance to tetracycline, beta-lactam, and multidrug. As Himalayan vultures were not exposed to antibiotics directly, the origin of these ARGs might be related to their feeding and living habits. Many yaks and Tibetan sheep raised on the Qinghai-Tibetan Plateau have become the main food source for the Himalayan vultures (*Lu et al., 2009*). Although data on the usage of tetracycline and beta-lactam in Qinghai-Tibetan Plateau are lacking, studies have established that tetracycline and beta-lactam are both the most extensively used antibiotic classes in most food-producing animals, including cattle, sheep, and poultry (*Economou & Gousia, 2015*). There is a high chance that Himalayan vultures may be exposed to residual antibiotics and ARB when feeding on these dead plateau livestock. Furthermore, livestock on the Qinghai-Tibetan Plateau are raised free-range, and antibiotics, which are not absorbed by the animals' intestines, are discharged into the surrounding environment with feces and urine (*Zubair et al., 2023*). Thus, wildlife inhabiting the same area will likely gain ARB and ARGs by sharing water or preying on food chains. Water bodies are also thought to cause the spread of ARGs in wildlife (*Jarma et al., 2021*). For the zoo animal populations, evidence of ARB and ARGs was also described in many captive species, making them reservoirs of ARGs (*Rodrigues da Costa & Diana, 2022*; *Min et al., 2023*). Compared with wild Himalayan vultures, captive individuals had more frequent contact with humans, which may account for the origin of ARGs in the zoo group. The food of the Himalayan vultures in the zoo mainly consisted of chicken (*Gallus gallus domesticus*), rabbit (*Oryctolagus cuniculus*), beef (*Bos taurus*), and lamb (*Ovis aries*), which were purchased from the market. This could be another way that captive Himalayan vultures acquire ARGs. Different comparative conclusions have been drawn between wild

and captive ARG levels in other species studied. For example, the fecal microbiomes of the captive sika deer (*Cervus nippon*) had more abundant ARGs than the free-range sika deer (*Wu et al., 2021*). The number and subtypes of ARGs were found to be higher in the feces of wild Asian elephants (*Elephas maximus*) than in those of captive elephants (*Cao et al., 2023*). Our results indicated that the percentage of shared ARGs was very high (up to 70%) between the wild and zoo groups, and ARG levels in captive Himalayan vultures were close to those of the wild ones based on beta diversity. This result may be due to the circulation of food resources on the Qinghai-Tibetan Plateau, especially the trade and circulation of beef and lamb, in which the dead cattle and sheep in the wild become the diet of wild Himalayan vultures, while the slaughtered cattle and sheep have become the diet of the captive individuals. These results partially confirmed that food sources may be the main determinant in forming Himalayan vulture ARGs (*Lee et al., 2022*).

Genes conferring resistance to tetracycline and beta-lactam multidrug in Himalayan vultures were dominant and common, consistent with studies reporting these genes in human-impacted soils and sewage (*Tang et al., 2023*). For instance, tetracycline is a broad-spectrum antimicrobial agent with low toxicity and is therefore widely used in treating human and animal infections, including as a growth promoter in food for livestock, poultry, and aquaculture, resulting in widespread tetracycline resistance. The most abundant tetracycline resistance (*tet*) gene was *tetP* in Himalayan vultures, involved in pumping the drug out of the cell before it reaches its site of action. The *tetP* genes were found in *Clostridium* spp., especially *Clostridium perfringens*, a zoonotic pathogen that causes infections in poultry, cows, and sheep (*Sloan et al., 1994*). We found that *Clostridium* was abundant in the gut microbiome of Himalayan vultures. Beta-lactam is among the most commonly prescribed drug classes with numerous clinical indications. The most abundant beta-lactam resistance gene in Himalayan vultures was *OXA-85*, involved in an oxacillinase that hydrolyzes the beta-lactam ring, rendering the antibiotic ineffective. The *OXA-85* gene was detected from the anaerobic gram-negative rod *Fusobacterium nucleatum* subsp. *polymorphum* (*Voha et al., 2006*). *Fusobacterium* was the dominant abundant genera in the gut microbiome of Himalayan vultures. Bacterial multidrug resistance continues to spread alarmingly, threatening human health worldwide. There are increasing reports of multidrug-resistant gram-negative infections in fecal samples of domestic animals (pigs, dogs, and horses) (*Szmolka & Nagy, 2013*). The most abundant multidrug resistance gene in Himalayan vultures was *acrB*. Motivated by the proton motive force, AcrB transfers substrates, including antibiotics, from the intracellular environment to the outside of the cell through a sealed channel formed by AcrA and TolC (*Du et al., 2014*). These findings suggest that the spread of *tetP*, *OXA-85*, and *acrB* genes between Himalayan vultures and domestic animals is of concern. Genes conferring resistance to MLS, bacitracin, and aminoglycoside were also common in Himalayan vultures. For instance, bacitracin was considered an inherent genetic trait prevalent in bacteria, as 153 genera of its homologs were found in the ARGs database (*Hu et al., 2013*). Furthermore, bacitracin occurs widely in reservoirs, household drinking water, and rivers, indicating that water is a potential route for the spread of these antibiotics.

Efflux pump and antibiotic deactivation were found to be the main resistance mechanisms of ARGs in Himalayan vultures, which might be partly due to the higher abundances of tetracycline resistance genes in Himalayan vultures (*Di Francesco et al., 2023*). Efflux pumps are ubiquitous in bacteria and can export various chemical compounds, reducing intracellular antibiotic concentration and conferring drug resistance (*Sun, Deng & Yan, 2014*). Moreover, the higher efflux pump abundance facilitates more bacteria adaptation to various environments, which makes it easier for ARGs to transfer horizontally between different bacterial hosts (*Zampieri, 2021*). Antibiotic deactivation was the second ARG resistance mechanism in Himalayan vultures, acted by enzymatic destruction or removal of antibiotics. For example, macrolide resistance is predominantly mediated through esterases or phosphotransferases (*Kakoullis et al., 2021*). Thus, different ARBs adopt different strategies to cope with antibiotic pressure. Further, the hosts of ARGs in Himalayan vultures were analyzed. We identified 75 genera (five phyla) of bacteria for the hosts of ARGs in Himalayan vultures, with *Fusobacterium* carrying beta-lactam, *Cetobacterium* and *Clostridium* carrying tetracycline, and *Escherichia* carrying multidrug. Because of the scavenging diet of vulture species, the gut microbiota was occupied by many pathogenic bacteria (*Roggenbuck et al., 2014*), so the host of the ARGs was also dominated by pathogenic bacteria. When there was antibiotic selection pressure in the food or environment, the gut pathogens of these Himalayan vultures were bound to produce ARGs, which could pose a public safety problem when these ARGs-carrying pathogens are transmitted to humans or other animals. Therefore, in the future, it is necessary to use culture-based methods to isolate and obtain gut pathogenic strains of Himalayan vultures, further study the evolution of ARGs, and screen antibiotic types that can kill pathogenic bacteria. The gut microbial structure of 41 phyla (1,451 genera) of Himalayan vultures identified using metagenomic sequencing provides a good foundation for the isolation and culture of gut microbes. Furthermore, we found that captivity and fixed diets did not affect the structure of captive Himalayan vultures' gut microbes as the gut microbiota of Himalayan vultures is selected by the strong immune system and gastric acid secretion, resulting in only specific gut bacterial being able to colonize the gastrointestinal region (*Zou et al., 2021*). Most of the gut commensal bacteria of Himalayan vultures were pathogenic to other animals; hence, they could be used as a biological sentinel species for pathogen surveillance in the future. Based on this, we analyzed the clinical and high-risk ARGs of Himalayan vultures. A total of 102 clinically relevant ARGs were identified in Himalayan vultures. If these clinically relevant ARGs combine with clinical pathogens, they will pose a significant threat to human health. For instance, the *tetQ* gene, encoding for a ribosomal protection protein conferring resistance to tetracycline, was found to be the dominant clinical ARG in Himalayan vultures. Through isolation methods, the most likely reservoirs of *tetQ* in chicken were found to be bacterial from phylum Bacteroidetes (*Juricova et al., 2021*). The association between clinical ARGs and clinical pathogens requires the cultured methods for further validation. The risk of ARGs to human health varies based on many factors, including host pathogenicity, genetic background, and the potential for transferring to human pathogens. Only a small proportion of the ARGs posed a threat to human health (*Fitzpatrick & Walsh, 2016*). Thus, identifying high-risk resistance genes

among many ARGs can save costs. Our study detected high-risk ARGs (35 Rank I and 56 Rank II ARGs) in Himalayan vultures. These ARGs are of concern and may have the highest risk of producing new or multidrug resistance in pathogens at present (Rank I-already present in pathogens) and in the future (Rank II- not yet present in pathogens).

MGEs facilitate the transfer of ARGs among different microorganisms in the environment (*Zhu et al., 2019*). Our study analyzed MGEs to assess their potential role in disseminating ARGs through HGT (Horizontal Gene Transfer). Five types of MGEs (128 subtypes) were identified in Himalayan vultures. The number of MGEs was much higher in the captive Himalayan vultures than in wild individuals, indicating a higher probability of dissemination of ARGs in captive individuals. This higher number of MGEs may be due to the high antibiotic pressure from intense human activity in zoos, whereas wild Himalayan vultures do not have close contact with people. The main MGE types in Himalayan vultures were plasmids and transposases, and their corresponding high-abundance subtypes were also found. For instance, there are high-abundance subtypes, such as *tnpA* and *IS91*. *tnpA* can catalyze the generation of the co-integrated structures in the replication of transposition (*Siguier, Gourbeyre & Chandler, 2017*). The *IS91* family transposase is typically related to multiple antibiotic resistance regions and thus may perform a key role in the dissemination and evolution of ARGs under antibiotic stress (*Toleman, Bennett & Walsh, 2006*). We then investigated the link among gut microbes, ARGs, and MGEs through the Procrustes and co-occurrence networks analysis. For the wild Himalayan vultures, Procrustes analysis revealed a significant correlation among gut microbes, ARGs, and MGEs, suggesting the gut microbial communities structured the gut ARGs and MGEs compositions. For the captive Himalayan vultures, Procrustes analysis indicated that ARGs correlated significantly only with MGEs but not with gut microbes, suggesting the gut microbial communities decoupled from the gut ARGs and MGEs compositions. This result may be because the food of the Himalayan vultures in the zoo is more fixed and simpler. By contrast, the diet of wild Himalayan vultures is more diverse, and ARGs and MGEs are more affected by the gut microbiota. Additionally, the co-occurrence networks analysis showed some highly correlated degree MGEs linked with most of the ARG subtypes and genera in both groups. Further studies are needed to determine whether ARGs associated with these highly correlated degree MGEs can switch hosts frequently within the gut microbial communities.

Overall, our study advances our understanding of the characteristics of the resistomes of Himalayan vultures and their changes between different environments. Future studies are needed to determine the influence of the emissions of ARGs on the environmental bacterial communities in this bird habitat, as well as to identify sources and sinks of ARGs related to this bird species. Furthermore, culture-based approaches should be used to study pathogenic and nonpathogenic strains to determine the resistance mechanisms of each bacterial species and their phylogenetic relationships with clinical ARGs. Functional metagenomics studies should be enhanced to discover unknown ARGs in future antimicrobial resistance studies of Himalayan vultures.

## CONCLUSION

Based on our study of the resistomes of Himalayan vultures, the scavenging birds carry large numbers of ARGs in the habitat environments where they feed and live. Although its effect on the recipient environment remains unknown, the specific genetic characteristics of the antibiotic resistance carried by birds remind us that the role of scavenging birds in the environmental transmission of ARGs cannot be ignored and merits more consideration.

### Funding

This research was funded by the National Natural Science Foundation of China (grant No. 31960277), and the Program of Science and Technology International Cooperation Project of Qinghai province (grant No. 2022-HZ-812). The funders had no role in study design, data collection and analysis, decision to publish, or preparation of the manuscript.

### Grant Disclosures

The following grant information was disclosed by the authors:
National Natural Science Foundation of China: 31960277.
Program of Science and Technology International Cooperation Project of Qinghai province: 2022-HZ-812.

### Competing Interests

The authors declare there are no competing interests.

### Author Contributions

- Jundie Zhai conceived and designed the experiments, performed the experiments, analyzed the data, prepared figures and/or tables, authored or reviewed drafts of the article, and approved the final draft.
- You Wang conceived and designed the experiments, performed the experiments, analyzed the data, prepared figures and/or tables, and approved the final draft.
- Boyu Tang conceived and designed the experiments, performed the experiments, analyzed the data, prepared figures and/or tables, and approved the final draft.
- Sisi Zheng conceived and designed the experiments, performed the experiments, analyzed the data, prepared figures and/or tables, and approved the final draft.
- Shunfu He conceived and designed the experiments, performed the experiments, analyzed the data, prepared figures and/or tables, and approved the final draft.
- Wenxin Zhao conceived and designed the experiments, performed the experiments, analyzed the data, prepared figures and/or tables, and approved the final draft.
- Jun Lin conceived and designed the experiments, performed the experiments, analyzed the data, prepared figures and/or tables, and approved the final draft.
- Feng Li conceived and designed the experiments, performed the experiments, analyzed the data, prepared figures and/or tables, and approved the final draft.

- Yuzi Bao conceived and designed the experiments, performed the experiments, analyzed the data, prepared figures and/or tables, and approved the final draft.
- Zhuoma Lancuo conceived and designed the experiments, performed the experiments, analyzed the data, prepared figures and/or tables, and approved the final draft.
- Chuanfa Liu conceived and designed the experiments, performed the experiments, analyzed the data, prepared figures and/or tables, authored or reviewed drafts of the article, and approved the final draft.
- Wen Wang conceived and designed the experiments, performed the experiments, analyzed the data, prepared figures and/or tables, authored or reviewed drafts of the article, and approved the final draft.

## Animal Ethics

The following information was supplied relating to ethical approvals (i.e., approving body and any reference numbers):

This study conformed to the guidelines for the care and use of experimental animals established by the Ministry of Science and Technology of the People's Republic of China (Approval number: 2006-398). The research protocol was reviewed and approved by the Ethical Committee of Qinghai University. This study did not involve capture or any direct manipulation or disturbance of Himalayan vultures.

## Data Availability

The raw sequence data are available at the Genome Sequence Archive in National Genomics Data Center, China National Center for Bioinformation/Beijing Institute of Genomics, Chinese Academy of Sciences (GSA): CRA012991.

## Supplemental Information

Supplemental information for this article can be found online at http://dx.doi.org/10.7717/peerj.17710#supplemental-information.

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
