# Peer review of "A comparison of antibiotic resistance genes and mobile genetic elements in wild and captive Himalayan vultures"

_PeerJ, doi:10.7717/peerj.17710_

## Round 0.1 · original submission · Major Revisions

· Academic Editor

Major Revisions

This manuscript has been reviewed by two experts. Both agree the importance of the current study contributing to the field. However, both raised some questions and suggestions for further improving the current form of the manuscript. Further revisions are required to address those concerns commented by reviewers.

Reviewer 1 ·

Basic reporting

no comment

Experimental design

no comment

Validity of the findings

no comment

Additional comments

The authors compared the ARGs and MGEs of gut microbiota in both wild and captive Himalayan vulture populations. The results are very useful for keeping an eye on this special group of birds. The specific characteristics of the antibiotic resistance carried by vulture birds remind us that the role of scavenging birds in the environmental transmission of ARGs cannot be ignored and merits more consideration. Here's a question I'm interested in. The results showed that the resistance genes and mobile elements were increased in captivity, and whether the dispersal of resistance genes should also be considered in the environmental release activities of some wild birds after rescue or artificial breeding, especially specific habitat. Additionally, I have some minor comments as follows.

Abstract:
Lines 37-44, Most of these descriptions are general results, and the presentation of differences between wild and captive populations may be a better summary and more appealing to the reader.

Introduction section:
Lines 83-85, This sentence needs to be revised.
Lines 86-102, Because of its special habitat, the Qinghai-Tibet Plateau is relatively less disturbed by human beings, and the significance of this study on antibiotic resistance of its endemic species need to be present.

Materials and Methods section:
Lines 111-112, How do you make sure that the fecal samples you collect comes from Himalayan vultures and not from another species?
Lines 145-147, What does the phrase "A read was labeled as an MGE, including" mean? Did you want to express “MGEs were obtained by labeling the read sequences, …”? If so, please restate the sentence.
Lines 179-180, "P < 0.5" ?
Line 181-185, “2.9. Data availability statement” This section repeats the part of Data Availability.

Results section:
Lines 189-191,Is “the resistance genes of beta-lactam, multidrug, tetracycline, aminoglycoside,… … and quinolone” right ?
Lines 229-234, Please check the full text and put the genus names of the microorganisms in italics.
Lines 231, Please change to “the top five genera”.
Line 248, 252, Change “gut microbes” to “54/47 gut microbial genera/groups”

Discussion section:
Lines 292-305, This paragraph belongs to the content of the introduction, please write it briefly.
Line 410, Don't need to add double quotation marks to high-risk words.

Reviewer 2 ·

Basic reporting

This is a very well-written, well considered article that investigated the antibiotic resistance genes in the gut microbiota of Himalayan vultures, both in the zoo and in the wild. The work is logical and well organised, and the English is professional Literature is well referenced throughout the work.
The graphs and tables are clearly explained and provide the information clearly. Raw data has been shared, alongside several supplemental files.

Experimental design

This document fits clearly within the remit of PeerJ. The rationale for the study is well explained, though there could be a bit more clarity surrounding the direction. For example, in some sections of the text, there is a suggestion that antibiotic resistance in Himalayan vultures (in general) is the main focus, whereas in others the focus is on the comparison between wild and captive birds. A bit more clarity would help to improve this. Ethics are clearly explained and the ethical impact of the study is minimal as the focus is on faecal matter. The sampling measures are clearly explained, and analysis is clear throughout the document.

Validity of the findings

While other antibiotic resistance studies have been conducted for a range of species, the focus on the Himalayan vulture provides some novelty. The authors indicate that the selection of vultures as a sentinel species could have some applications to wild environments. The data are clearly explained throughout the work. The conclusions could be made a little clearer in the discussion- consider stating the key findings in the first instance.

Additional comments

This is a thoughtful and well formatted study that has some potential connotations in terms of ecological health. I have added just a few minor comments to address, which can be found on the PDF of the manuscript. Well done to the authors on this well-written manuscript.

Annotated reviews are not available for download in order to protect the identity of reviewers who chose to remain anonymous.

---

## Round 0.2 · accepted · Accept

· Academic Editor

Accept

This revised manuscript has been much improved. It also appropriately responded to each comment of two reviewers. It is ready for publication in PeerJ.